# Intravascular Ultrasound Findings in Acute and Chronic Deep Vein Thrombosis of the Lower Extremities

**DOI:** 10.3390/diagnostics15050577

**Published:** 2025-02-27

**Authors:** Fabio Corvino, Francesco Giurazza, Massimo Galia, Antonio Corvino, Roberto Minici, Antonio Basile, Anna Maria Ierardi, Paolo Marra, Raffaella Niola

**Affiliations:** 1Interventional Radiology Department, AORN “A. Cardarelli”, 80131 Naples, Italy; francesco.giurazza@aocardarelli.it (F.G.); raffaella.niola@aocardarelli.it (R.N.); 2Section of Radiology, Department of Biomedicine, Neuroscience and Advanced Diagnostics (BiND), University Hospital “Paolo Giaccone”, 90127 Palermo, Italy; massimo.galia@unipa.it; 3Medical, Movement and Wellbeing Sciences Department, University of Naples “Parthenope”, 80133 Naples, Italy; antonio.corvino@uniparthenope.it; 4Radiology Unit, University Hospital Dulbecco, 88100 Catanzaro, Italy; miniciroberto@gmail.com; 5Radiology Unit 1, Department of Medical Surgical Sciences and Advanced Technologies “GF Ingrassia”, University Hospital “Policlinico-San Marco”, University of Catania, 95123 Catania, Italy; basile.antonello73@gmail.com; 6Department of Diagnostic and Interventional Radiology, Foundation IRCCS Cà Granda-Ospedale Maggiore Policlinico, 20122 Milan, Italy; annamaria.ierardi@policlinico.mi.it; 7Department of Radiology, Papa Giovanni XXIII Hospital, 24127 Bergamo, Italy; pmarra@asst-pg23.it

**Keywords:** intravascular ultrasound (IVUS), thrombosis, deep vein thrombosis

## Abstract

Deep vein thrombosis (DVT) of the lower extremities, as part of venous thromboembolism disorder, is the third leading cause of acute cardiovascular syndrome after heart attack and stroke. It can result in disability due to pulmonary embolism (PE) and post-thrombotic syndrome (PTS), particularly in cases where the thrombosis extends to the iliofemoral veins. Anticoagulation therapy is effective in preventing thrombus propagation and embolism but may not be sufficient for thrombus degradation and venous patency restoration. Up to 50% of patients with iliofemoral DVT develop PTS, mainly due to venous outflow obstruction or valvular incompetence. To date, the advent of new devices that enables rapid thrombus elimination and the restoration of deep venous patency, known as the “OPEN VEIN hypothesis”, may prevent valvular damage and reflux, cutting down the rate of PTS. Similarly, chronic venous disease could be related to a stenosis or occlusion of a major vein that can restrict blood flow. In this setting, intravascular ultrasound (IVUS) is an essential tool for correct diagnostic and therapeutic planning in acute and chronic vein disease. Only angiography in vein disease can limit the procedure’s efficacy, with a high rate of stenosis misdiagnosed; IVUS provides further imaging that complements traditional angiographic study, and its role is now established by different international guidelines. If compared to angiography, IVUS allows for the evaluation of major axial veins in a 360-degree ultrasound image of the lumen and of the vessel wall structure. At the same time, the precise location and size of the major lower extremity veins allow for the placement of the stent to be more straightforward with a precise dimension of the vein in all of its diameters; moreover, other abnormalities should be visualized as acute or chronic thrombus, fibrosis, or trabeculations. This review aims to provide an in-depth analysis of IVUS findings in acute and chronic lower extremity DVT, emphasizing its diagnostic and therapeutic implications.

## 1. Introduction

Deep vein thrombosis (DVT) is a significant clinical condition associated with substantial morbidity and mortality, impacting millions annually worldwide. Complications such as post-thrombotic syndrome (PTS) and pulmonary embolism (PE) underscore the necessity for accurate diagnosis and timely management. Venous thromboembolism (VTE) continues to be a devastating problem, representing the third most common cause of death from cardiovascular disease. The approach to this pathological condition is changing to a more aggressive approach that includes not only the prevention but also the treatment in acute setting VTE [1,2].

With the increase in data that support VTE treatment in acute/subacute settings, standard devices that are used in arterial procedures are now also applied to VTE treatment. While traditional imaging modalities such as duplex ultrasound and venography remain widely utilized, these methods have limitations in resolution and depth assessment, particularly in complex or anatomically challenging cases [3].

Intravascular ultrasound (IVUS) has emerged as a superior imaging tool that provides high-resolution, real-time visualization of the venous lumen and wall. Unlike traditional approaches, IVUS enables a detailed assessment of thrombus composition, vein wall pathology, and adjacent structures. Furthermore, it facilitates procedural guidance during interventions such as catheter-directed thrombolysis and stent placement [4,5].

This review explores the key IVUS findings that characterize acute DVT and chronic changes that lead to PTS, along with their clinical implications and the role of IVUS in guiding treatment strategies. By providing a detailed comparison of acute versus chronic thrombotic changes, this review highlights the value of IVUS in enhancing diagnostic accuracy, informing interventional procedures, and improving patient outcomes.

## 2. Current Guidelines for the Management of Acute and Chronic Deep Vein Thrombosis (DVT)

The management of acute and chronic DVT has evolved significantly, with various international guidelines providing evidence-based recommendations to optimize patient outcomes. The European Society for Vascular Surgery (ESVS), the Society of Interventional Radiology (SIR), the Association of the Scientific Medical Societies in Germany (AWMF), and the American College of Chest Physicians (ACCP) have all emphasized a multidisciplinary approach that integrates anticoagulation therapy, endovascular interventions, and long-term follow-up [6,7,8,9]. Table 1 provides a comparative overview of the major international guidelines (ESVS, SIR, AWMF, and ACCP) regarding the management of acute and chronic DVT. It summarizes key recommendations on anticoagulation strategies, endovascular treatment approaches, and the role of IVUS in procedural planning and intervention optimization.

One of the fundamental aspects of DVT management is the initial diagnostic approach, which relies on clinical probability assessment, typically using the Wells score, in conjunction with D-dimer testing. This stratification allows clinicians to determine the likelihood of a thrombotic event and proceed with appropriate imaging. Venous duplex ultrasonography (VDUS) is universally recommended as the first-line imaging modality, particularly for lower extremity DVT, due to its non-invasiveness and high sensitivity. However, in cases where there is suspicion of iliofemoral or iliocaval involvement, alternative imaging techniques such as computed tomography venography (CTV) or magnetic resonance venography (MRV) are often required to obtain a more detailed anatomical assessment [9,10,11].

Among the more advanced imaging modalities, intravascular ultrasound (IVUS) has gained recognition as a superior tool for evaluating thrombus burden, vein wall characteristics, and the degree of luminal obstruction. While traditional imaging techniques provide a two-dimensional perspective, IVUS offers real-time, high-resolution cross-sectional imaging, which enhances the accuracy of procedural planning and execution in both acute and chronic venous thrombosis [4,5,6,7,8,9,10,11,12,13].

When it comes to the treatment of acute DVT, anticoagulation remains the cornerstone of therapy. Across all major guidelines, direct oral anticoagulants (DOACs) are preferred over vitamin K antagonists (VKAs) due to their favorable safety profile and convenience. The standard duration of anticoagulation therapy is three to six months, with extensions based on individual patient risk factors, such as recurrent thrombosis or persistent provoking conditions [14,15].

The ACCP guidelines emphasize the importance of risk stratification for anticoagulation therapy. They recommend a three-month treatment phase of anticoagulation for patients with acute VTE without contraindications. For patients with unprovoked VTE or persistent risk factors, extended-phase anticoagulation with DOACs is strongly recommended, while vitamin K antagonists (VKAs) are suggested for those who cannot take DOACs [9].

In patients with iliofemoral DVT, where the thrombus burden is significant and there is an increased risk of developing PTS, endovascular thrombus removal (ETR) techniques, such as catheter-directed thrombolysis (CDT) and mechanical thrombectomy (MT), may be warranted. These approaches aim to reduce thrombus volume and preserve venous function, thereby minimizing the long-term complications associated with chronic venous obstruction (CVO) [7,8,9,10,11,12,13,14,15,16].

For patients with PTS, the treatment strategy often involves venous stenting to restore proper blood flow and alleviate symptoms. The recently proposed CVO classification system provides a structured framework for categorizing patients based on the anatomical extent of obstruction, offering valuable prognostic insights. This classification distinguishes different types of iliofemoral CVO, helping to predict stent patency rates and optimize treatment strategies [17]. The ACCP guidelines also highlight that IVUS plays a crucial role in ensuring precise stent sizing and confirming adequate venous inflow, which is essential for achieving long-term procedural success. Although the benefits of IVUS in DVT management are well documented, its adoption remains inconsistent across clinical practices, largely due to cost considerations and the need for specialized training [18,19].

Some guidelines, such as those from the ESVS, recognize the utility of IVUS for anatomical assessment but do not mandate its use as standard practice. In contrast, the SIR and ACCP strongly advocate for IVUS guidance in procedural settings, particularly for iliofemoral interventions, where its ability to provide real-time insights significantly enhances technical outcomes [6,7].

## 3. IVUS in Acute Lower Extremity DVT

In acute DVT, IVUS serves as an essential imaging modality, offering real-time insights into thrombus morphology and associated venous wall changes. Compared to conventional imaging techniques, IVUS enables direct visualization of thrombotic structures, enhancing diagnostic accuracy and treatment planning [19,20,21,22,23].

Acute thrombi often present with a hypoechoic or isoechoic appearance, indicating their relatively soft and unorganized nature (Figure 1). This characteristic echotexture allows differentiation from chronic thrombi, typically more echogenic due to increased fibrosis and organization. IVUS also reveals whether the thrombus is causing a partial or complete occlusion of the venous lumen, a critical factor in determining therapeutic strategies. Unlike chronic thrombi, which exhibit well-defined, rigid contours, acute thrombi often have irregular, poorly demarcated edges, which can influence catheter-directed thrombolysis or mechanical thrombectomy approaches [21,22].

The impact of IVUS extends beyond thrombus characterization to the evaluation of venous wall integrity. In acute DVT, the affected vein frequently appears distended due to elevated intraluminal pressure from the obstructive thrombus, yet without significant wall thickening or fibrosis, which are hallmarks of chronic venous disease. This distinction is particularly relevant when determining eligibility for endovascular interventions as studies have shown IVUS-guided therapies, including catheter-directed thrombolysis, result in improved thrombus resolution and lower rates of residual clot burden compared to venography alone (Figure 2). Additionally, IVUS can help identify subtle wall abnormalities, such as early-stage fibrosis or endothelial damage, which may predict long-term complications such as PTS. Another key finding in acute DVT cases is the minimal or absent collateral vein formation. Unlike CVO, where collateral pathways develop over time to bypass occluded segments, acute thrombosis does not typically present with significant alternative venous drainage. This observation is crucial in differentiating acute from chronic disease and in assessing the severity of venous outflow obstruction. IVUS further enhances this evaluation by providing real-time hemodynamic assessments, including detecting regions with reduced or absent venous flow. When used in conjunction with Doppler ultrasound, IVUS enables precise localization of thrombotic occlusions [4,5,6,7,8,9,10,11,12,13,14,15,16,17,18,19,20,21,22,23].

*Clinical Implications.* The clinical utility of IVUS in acute DVT extends beyond diagnostics, playing a fundamental role in procedural guidance and post-intervention assessment. By offering direct visualization of thrombus composition, IVUS minimizes the diagnostic uncertainties associated with traditional imaging modalities such as duplex ultrasound or venography. Its ability to assess the extent of thrombus burden and vein wall morphology provides critical information for selecting optimal treatment approaches, whether pharmacological, mechanical, or combined strategies. Furthermore, IVUS has improved the accuracy of stent placement and thrombolytic catheter positioning, leading to enhanced clinical outcomes and reduced rates of re-thrombosis. In the setting of interventional procedures, IVUS-guided catheter-directed thrombolysis and mechanical thrombectomy offer superior thrombus resolution compared to conventional approaches [24,25].

## 4. IVUS in PTS

In PTS, IVUS reveals a distinct set of sonographic characteristics that reflect the prolonged nature of the disease process; unlike the soft, hypoechoic thrombi observed in acute DVT, chronic thrombi present as hyperechoic or heterogeneous structures, indicative of advanced fibrosis, calcification, and organization over time. These thrombi are frequently well-adherent to the venous endothelium, demonstrating firm, irregular, and well-defined borders contributing to persistent venous obstruction. The rigidity and dense structure of chronic thrombi reduce their responsiveness to thrombolytic therapy, necessitating alternative interventional strategies such as mechanical thrombectomy or stenting (Figure 3) [4,5,6,7,8,9,10,11,12,13,14,15,16,17,18,19,20,21,22,23,24,25,26].

IVUS provides a critical advantage in assessing the extent of luminal compromise in PTS cases. These thrombi often result in significant luminal narrowing, contributing to chronic venous hypertension. The vein wall in PTS patients undergoes substantial remodeling, including thickening, scarring, and fibrotic band formation, which can exacerbate symptoms and lead to long-term venous insufficiency. Unlike acute thrombi, which causes transient venous distension, chronic venous thrombosis is associated with progressive endothelial damage, leading to permanent venous dysfunction. IVUS enables real-time visualization of these pathological changes, facilitating differentiation between residual chronic thrombi and new thrombotic events (Figure 4) [3,4,5,6,7,8,9,10,11,12,13,14,15,16,17,18,19,20,21,22,23,24,25,26,27].

A hallmark feature of PTS is the extensive development of collateral venous circulation. Over time, the body compensates for persistent venous occlusion by forming alternative drainage pathways that attempt to maintain adequate venous return. IVUS effectively maps these collateral networks, distinguishing functional compensatory channels from insufficient or varicose pathways that may contribute to venous reflux (Figure 5). Additionally, IVUS frequently detects valvular dysfunction, a common consequence of chronic thrombotic disease. Prolonged venous hypertension and fibrotic changes can lead to valve destruction or incompetence, resulting in significant reflux and impaired venous hemodynamics. These hemodynamic alterations increase the risk of chronic venous insufficiency, edema, and venous ulceration, particularly in patients with a history of recurrent thrombotic episodes [28]. Table 2 summarizes the main differences in IVUS findings between acute DVT and PTS syndrome.

*Clinical Implications.* IVUS plays a pivotal role in distinguishing chronic thrombotic disease from acute exacerbations, a distinction that has critical implications for treatment planning. By accurately identifying residual fibrotic thrombi, IVUS prevents unnecessary anticoagulation or the inappropriate use of thrombolytic agents, which may be ineffective in well-organized, calcified thrombi. This differentiation is particularly relevant for patients with a history of multiple thrombotic episodes, where clinical symptoms alone may not provide sufficient clarity regarding disease chronicity [17,18,19,20,21,22,23,24,25,26,27,28,29].

In the interventional setting, IVUS is indispensable for guiding endovascular procedures such as angioplasty and stenting. By providing precise measurements of luminal diameter and detecting the extent of fibrotic occlusion, IVUS ensures optimal stent sizing and placement. This is particularly crucial in conditions such as May-Thurner syndrome, where chronic iliac vein compression and scarring necessitate strategic intervention [30]. The use of IVUS in these procedures has been shown to improve stent patency rates and reduce the incidence of restenosis by allowing for meticulous lesion characterization and targeted treatment. Furthermore, IVUS facilitates comprehensive post-intervention assessment, ensuring adequate stent expansion and detecting residual flow-limiting abnormalities. Studies have demonstrated that IVUS-guided venous interventions lead to superior long-term clinical outcomes, reducing the incidence of re-thrombosis and improving overall venous hemodynamics [30]. Given its ability to enhance procedural precision, optimize therapeutic strategies, and improve patient prognosis, IVUS is now considered an essential imaging modality in the management of chronic lower extremity DVT [31].

## 5. Advantages of IVUS over Traditional Imaging

IVUS presents numerous advantages over traditional imaging modalities, particularly in the context of DVT diagnosis and treatment planning. Conventional imaging techniques, such as duplex ultrasound, CTV, and MRV, each have inherent limitations that IVUS overcomes [4,5,6,7,8,9,10,11,12,13,14,15,16,17,18,19,20,21,22,23,24,25,26,27,28,29,30,31,32,33,34].

One of the primary benefits of IVUS is its ability to provide real-time, high-resolution, cross-sectional images of the vessel lumen and wall. Unlike traditional venography, which offers only a two-dimensional projection and is limited in detecting subtle vein wall abnormalities, IVUS enables direct visualization of thrombus composition, residual luminal diameter, and wall morphology. This is particularly useful in differentiating between acute and chronic thrombotic conditions guiding treatment decisions such as thrombolysis or mechanical thrombectomy [3].

Additionally, IVUS excels in identifying non-thrombotic iliac vein lesions (NIVL), such as May-Thurner syndrome, which may be missed or underestimated with standard venographic techniques. It allows for precise measurement of venous stenosis, which is crucial for selecting appropriate stent sizes and optimizing placement. Studies have shown that IVUS-guided interventions result in superior outcomes compared to procedures relying solely on venography. Toh et al., comparing IVUS with CTV, demonstrated a lower sensibility for CTV in the significant stenosis identification, suggesting subjecting the patients with a normal CTV and ilio-femoral clinical symptoms also to an IVUS study [35].

When comparing IVUS to other imaging modalities for assessing stenosis in patients with acute or chronic iliofemoral thrombosis, several distinctions emerge. Traditional venography, despite being widely used, has been found to underestimate the severity of venous stenosis. Studies have shown that venography fails to detect stenotic lesions in up to 19% of cases and often misidentifies the location of maximal stenosis [36].

Similarly, multiplanar venography, while slightly more accurate than single-view venography, still lacks the sensitivity of IVUS, particularly in cases of PTS [37,38].

CTV provides valuable anatomical insights and can help with pre-procedural planning but lacks real-time assessment capabilities. A study comparing CTV and IVUS demonstrated that CTV had a sensitivity of 94% in detecting iliac vein obstruction but tended to overestimate luminal areas compared to IVUS [39,40].

MRV, though non-invasive, has shown lower specificity in detecting clinically significant venous obstructions and is often limited by motion artifacts and the need for contrast administration [41].

A recent 2022 study comparing IVUS with MRV further demonstrated that IVUS provides superior visualization of venous lesions compared to MRV. IVUS exhibited higher sensitivity in detecting CVO, offering more precise information on stenosis morphology and severity. These findings suggest that IVUS may be preferable to MRV for a detailed assessment of chronic iliac vein obstructions [42].

Multiple studies have compared IVUS with other imaging modalities for detecting venous stenosis and thrombus burden. A meta-analysis reported that IVUS has a sensitivity of 92–98% and a specificity of 95–99%, outperforming traditional venography, which has a sensitivity of 90–96% and a specificity of 87–95%. Similarly, duplex ultrasound, while non-invasive, has a lower sensitivity (80–95%) for detecting iliac vein stenosis, particularly in obese patients or cases of chronic thrombosis. CTV and MRV have higher sensitivity (85–97%) but lack the real-time procedural guidance that IVUS provides [3]. Table 3 comprehensively compares IVUS with other imaging modalities in assessing venous disease.

The VIDIO trial, a multicenter study, highlighted that IVUS has shown significantly higher sensitivity than venography, with studies reporting a sensitivity of 92–98% compared to 90–96% for venography. IVUS identified stenotic lesions in 81% of cases, whereas venography detected them in only 51%. Additionally, IVUS provided more accurate measurements of stenotic severity, directly influencing treatment decisions and leading to improved clinical outcomes [43]. 

Given these comparative advantages, IVUS is increasingly recognized as the gold standard for diagnosing and guiding interventions in iliofemoral venous disease. Its superior sensitivity, ability to assess vessel wall morphology, and real-time imaging capabilities make it indispensable for optimizing patient outcomes in both acute and chronic settings.

## 6. IVUS in Venous Stenting Procedure

Achieving precise and effective stent placement is paramount for optimizing venous flow restoration and minimizing the risk of in-stent restenosis or thrombosis. IVUS plays a crucial role in this process by allowing accurate measurement of venous lumen dimensions, which is fundamental in selecting the most appropriate stent diameter. Correct stent sizing ensures complete luminal coverage while preventing complications associated with oversizing, such as migration, malpositioning, or extrinsic compression. The ultimate goal of venous stenting is to alleviate venous hypertension, decompress the peripheral venous network, and improve overall hemodynamics. Restoring adequate venous inflow dictates long-term stent patency and prevents recurrent occlusion (Figure 6) [5,6,7,8,9,10,11,12,13,14,15,16,17,18,19,20,21,22,23,24,25,26,27,28,29,30,31,32,33,34,35,36,37,38,39,40,41,42,43,44].

Unlike the arterial system, where the perfusion is driven by a constant cardiac output, the venous system is directly influenced by peripheral venous return and hemodynamic variations. This dynamic nature necessitates a precise stent caliber selection that accommodates physiological inflow without causing iatrogenic stenosis or residual obstruction. IVUS enhances procedural success by providing real-time assessment of vessel diameter, ensuring the selected stent aligns with physiological inflow demands. Undersizing a venous stent can lead to residual stenosis, which may compromise blood flow and increase the likelihood of reintervention [29,30,31,32,33,34,35,36,37,38,39,40,41,42,43,44,45,46].

IVUS is particularly valuable during deployment, enabling real-time monitoring of stent expansion and apposition against the vein wall. Post-deployment IVUS evaluations help detect residual stenosis, malpositioning, or complications, such as intimal hyperplasia, which can negatively impact long-term stent performance. Clinical studies have demonstrated that IVUS-guided stent placement leads to superior mid-term and long-term patency rates compared to venography alone, significantly reducing the likelihood of reintervention [32,33,34,35,36,37,38,39,40,41,42,43,44,45,46,47].

In clinical practice, the ability to confirm optimal stent positioning intraoperatively has been associated with enhanced patient outcomes and decreased rates of restenosis or occlusion. The evidence underscores the importance of IVUS in venous stenting, not only for procedural accuracy but also for its role in optimizing long-term venous hemodynamics. Given its capacity to refine stent selection, improve placement precision, and mitigate procedural complications, IVUS is widely regarded as a critical tool in modern venous interventions [21].

The degree of stenosis measured by IVUS is crucial in determining the need for intervention, especially in cases where traditional imaging modalities, such as venography, duplex ultrasound, or CTV, fail to detect significant stenosis. Unlike venography, which can underestimate luminal narrowing due to its reliance on contrast filling, IVUS provides a cross-sectional, real-time assessment of the vessel wall and luminal area [3].

Studies have demonstrated that stenosis ≥ 50% on IVUS often justifies intervention, particularly in symptomatic patients with PTS. The VIDIO trial found that IVUS detected stenoses ≥ 50% in 81% of patients with negative findings on venography, leading to improved clinical outcomes after stenting. Additionally, a luminal area reduction of >75% on IVUS strongly correlates with symptomatic venous outflow obstruction, even when venography or CTV appears normal [43].

The discrepancy between IVUS and traditional imaging is particularly evident in iliac vein compression syndromes (e.g., May-Thurner Syndrome), where IVUS often reveals more significant narrowing than venography. This has led to its increasing adoption as the gold standard for venous assessment, ensuring that patients with occult but clinically significant stenosis receive appropriate treatment [39,40].

IVUS is also essential for determining the optimal stent diameter, as it measures the native vein’s cross-sectional area rather than relying solely on angiographic dimensions. This approach minimizes the risk of stent undersizing or oversizing, major contributors to restenosis and stent migration [4,5,6,7,8,9,10,11,12,13,14,15,16,17,18,19,20,21,22,23,24,25,26,27,28,29,30]. Table 4 summarizes the role of IVUS in procedural guidance, highlighting its advantages, alternative imaging methods, and the associated risks when IVUS is not used.

Overall, IVUS provides both anatomical and functional insights that significantly impact venous intervention decision-making. Future research should aim to standardize IVUS-based criteria for intervention, ensuring that patients with hemodynamically significant but angiographically occult stenoses receive appropriate treatment [20].

## 7. Technical Advances and Emerging Applications

The continuous evolution of IVUS technology has significantly broadened its clinical utility, particularly with the advent of next-generation systems featuring higher-frequency transducers and improved resolution. These advancements allow for more precise visualization of vascular structures, enhancing the detection of subtle venous abnormalities that may not be fully appreciated with traditional imaging techniques. Modern IVUS platforms offer hybrid capabilities, integrating fluoroscopic imaging with three-dimensional reconstructions to optimize procedural accuracy. Furthermore, artificial intelligence (AI)—assisted IVUS systems are being actively developed to refine image acquisition, reduce operator variability, and facilitate automated interpretation of venous pathology. Integrating AI-driven algorithms with IVUS can improve diagnostic efficiency by enabling real-time pattern recognition, thus assisting clinicians in differentiating acute from chronic thrombotic disease and predicting stent patency outcomes [48,49].

Beyond its established role in DVT management, IVUS is gaining traction in a wider range of venous disorders. It has proven instrumental in assessing venous compressive syndromes such as May-Thurner syndrome, nutcracker syndrome, and NIVLs, where precise anatomical mapping is critical for determining the need for stenting. Unlike conventional multiplanar venography, IVUS enables direct visualization of luminal stenosis and vessel wall compression, improving patient selection and tailored intervention strategies. Recent studies indicate that IVUS-guided interventions in iliac vein compression have led to better post-procedural outcomes, with reduced rates of restenosis and improved long-term venous patency compared to venography-guided approaches [20,21,22,23,24,25,26,27,28,29,30,31,32,33,34,35,36,37,38,39,40,41,42,43,44,45,46,47,48,49,50].

Despite these advancements, IVUS is not without limitations. The technology remains costly, requiring specialized equipment and highly trained operators to ensure optimal image interpretation and procedural execution. Its invasive nature compared to noninvasive imaging modalities such as CTV or MRV presents an additional challenge, particularly in settings where procedural efficiency and cost-effectiveness are primary concerns. Nevertheless, ongoing innovations in IVUS technology, including catheter miniaturization and enhanced Doppler-based flow assessment, aim to address these barriers and expand its accessibility. The integration of machine learning algorithms into IVUS workflows holds significant promise for automating key aspects of image analysis, potentially reducing inter-operator variability and improving diagnostic accuracy. By leveraging these advancements, IVUS is poised to play an increasingly pivotal role in the endovascular management of complex venous disease [21,22,23,24,25,26,27,28,29,30,31,32,33,34,35,36,37,38,39,40,41,42,43,44,45,46,47,48,49,50,51].

## 8. Discussion

The growing body of evidence supporting IVUS as a superior diagnostic and interventional tool in venous disease management is compelling. Compared to traditional imaging modalities such as venography, CTV, and MRV, IVUS has consistently demonstrated superior diagnostic accuracy, procedural efficacy, and long-term patient outcomes. Its ability to provide real-time, high-resolution cross-sectional imaging of venous structures allows for precise assessment of thrombus morphology, vein wall characteristics, and luminal dimensions, ultimately leading to more informed therapeutic decision-making. Furthermore, IVUS has shown substantial advantages in guiding venous interventions, particularly in stent placement for CVO, where it improves stent sizing, placement precision, and long-term patency [11,12,13,14,15,16,17,18,19,20,21,22].

Despite these advantages, IVUS adoption remains inconsistent across clinical practices. The primary barriers include high equipment costs, the need for specialized training, and the perception of increased procedural complexity. Additionally, while studies have demonstrated its clinical benefits, further research is needed to establish cost-effectiveness and long-term superiority over conventional imaging approaches. Large-scale, randomized controlled trials comparing IVUS-guided interventions to standard venography-based techniques across diverse patient populations would provide more definitive evidence to position IVUS as the gold standard in venous imaging. Standardized treatment protocols incorporating IVUS into routine VTE management could also facilitate broader clinical adoption and streamline best practices in interventional radiology [23,24,25,26,27,28,29,30,31,32,33,34,35,36,37,38,39,40,41,42,43,44].

The expanding applications of IVUS beyond DVT further underscore its transformative impact on vascular imaging. Its role in diagnosing and managing venous compressive syndromes, such as May-Thurner and Nutcracker syndrome, is well documented, offering precise anatomical mapping and real-time assessment of luminal stenosis. Additionally, IVUS is becoming increasingly valuable in evaluating venous outflow obstruction in patients with PTS, where it aids in distinguishing residual thrombus from fibrotic stenosis, guiding optimal stenting strategies. Emerging technological enhancements, including hybrid IVUS-fluoroscopy systems and AI-driven image analysis, promise to optimize its clinical utility by reducing operator dependency, enhancing diagnostic accuracy, and improving procedural efficiency [15,16,17,18,19,20].

Future advancements in AI and catheter miniaturization will further refine IVUS applications, enhancing diagnostic precision and procedural efficiency while reducing operator dependency. These innovations will make IVUS an increasingly integral part of VTE management. With its superior diagnostic capabilities, IVUS is emerging as a key tool for standardizing venous interventions, particularly in iliofemoral stenting, as recognized in the CVO classification system [17].

In conclusion, IVUS is redefining the approach to acute and chronic DVT by offering unmatched precision in thrombus characterization, vein wall remodeling assessment, and procedural guidance. As research continues to refine its applications and standardize its integration into clinical practice, IVUS is poised to become a cornerstone in venous imaging and intervention. While challenges related to cost and training persist, the undeniable clinical benefits of IVUS, coupled with ongoing technological advancements, support its increasing adoption in modern interventional radiology. As further research refines treatment protocols and solidifies IVUS-guided approaches, this technology is well-positioned to become an essential component of comprehensive DVT care.

## Figures and Tables

**Figure 1 diagnostics-15-00577-f001:**
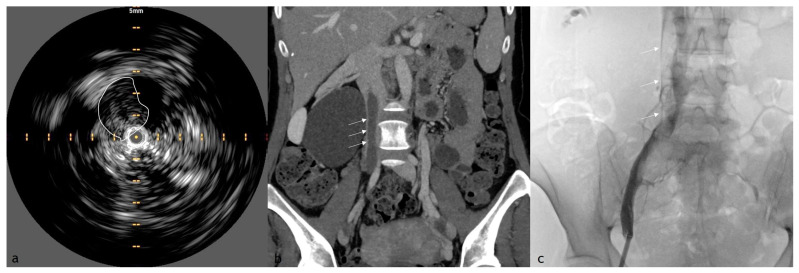
IVUS, CTV, and XA comparison images of a 55-year-old woman presented to our ED for breath shortness and pulmonary embolia who underwent a mechanical thrombectomy procedure. (**a**) IVUS cross-sectional image showing a hypoechoic thrombus (outlined) within the vein lumen; (**b**) CTV that shows the partially occlusive thrombus in the right iliac vein extended to inferior vena cava; (**c**) XA depicting intraluminal filling defects (arrows) corresponding to the thrombus seen on IVUS and CTV.

**Figure 2 diagnostics-15-00577-f002:**
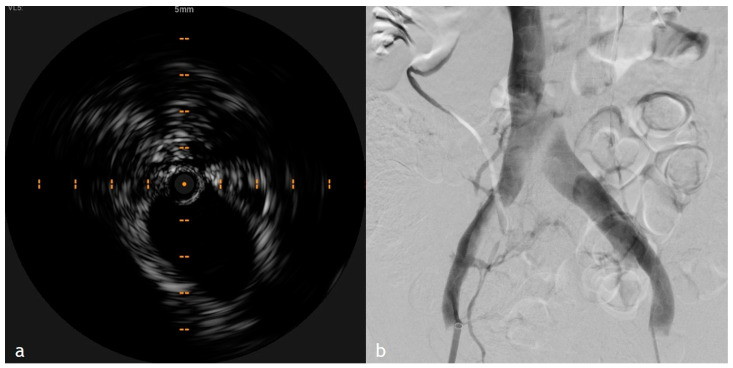
Comparison of IVUS and XA images after mechanical thrombectomy procedure. (**a**) IVUS image showed the complete flow restoration with no evidence of residual thrombus; (**b**) XA final control that showed an incomplete opacification of the inferior vena cava related to flow because the IVUS demonstrated no more presence of opacification defect with complete flow restoration. This comparative imaging highlights the importance of IVUS in complementing traditional venography for precise diagnosis and procedural guidance in venous interventions.

**Figure 3 diagnostics-15-00577-f003:**
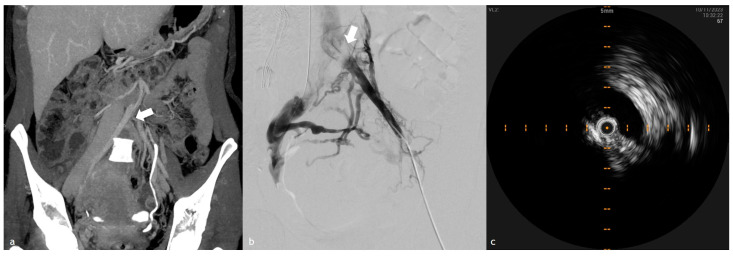
Multimodal imaging of chronic iliofemoral venous occlusion with collateral circulation. (**a**) CTV coronal reconstruction highlighting an occluded iliac vein with extensive collateral venous circulation (white arrow), indicative of chronic venous outflow obstruction. (**b**) Venography demonstrates an extensive collateral veins pathway bypassing the occlusion (white arrow), confirming hemodynamic impairment. (**c**) Cross-sectional IVUS image showing a narrowed and fibrotic vessel lumen, confirming the chronic nature of the venous obstruction.

**Figure 4 diagnostics-15-00577-f004:**
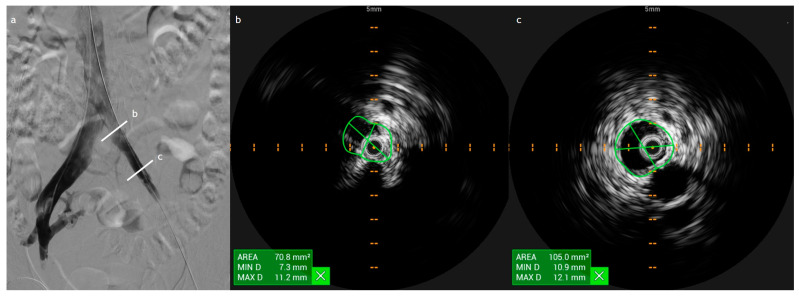
The same patient in Figure 3 after the PTA procedure. Fluoroscopic and IVUS assessment of iliac vein stenosis after PTA. (**a**) Contrast injection demonstrating residual iliac vein stenosis (marked by lines at points b), with a reduced luminal diameter and a healthy vein segment (marked by lines at points c) that corresponds to the iliac vein stent landing zone; (**b**,**c**) Cross-sectional IVUS image showing the narrowed lumen and the area measurement, essential value for the stenting procedure.

**Figure 5 diagnostics-15-00577-f005:**
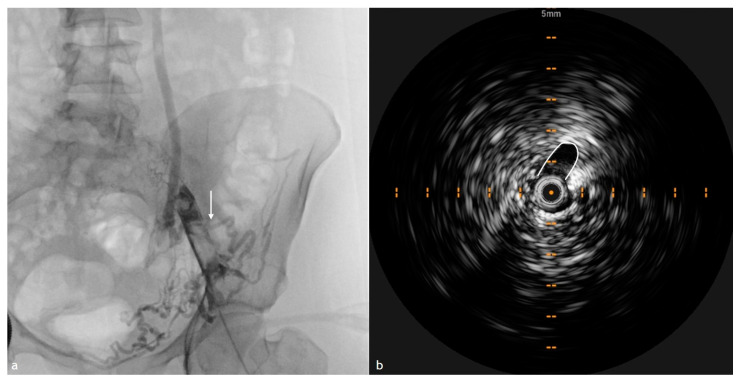
Collateral venous circulation in chronic venous occlusion: fluoroscopic venography and IVUS Comparison. (**a**) Fluoroscopic venography: The contrast injection highlights the presence of collateral venous circulation (arrow), indicative of chronic venous obstruction and compensatory drainage pathways with ovarian and vein opacification. (**b**) The cross-sectional IVUS image demonstrates intraluminal stenosis with the patent collateral pathway, identified in the previous image with an arrow, outlined. This multimodal imaging approach emphasizes the role of IVUS in providing a detailed assessment of vessel lumen morphology, complementing traditional venography in the evaluation of chronic venous disease.

**Figure 6 diagnostics-15-00577-f006:**
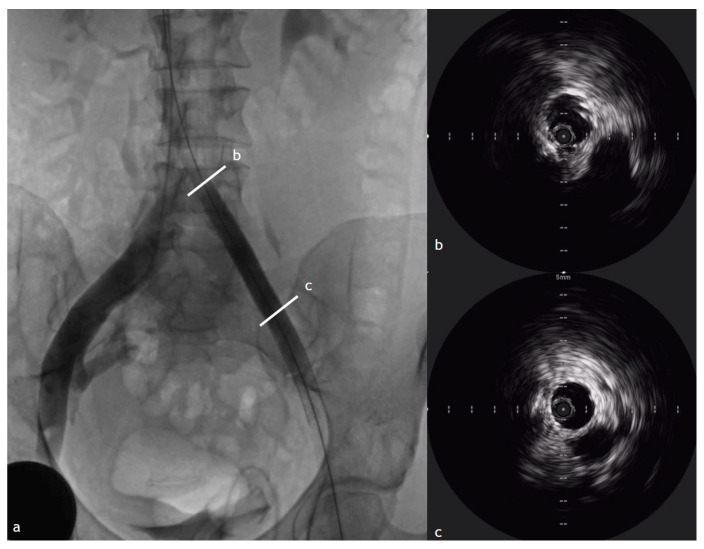
The same patient in Figure 4 after the stenting procedure. (**a**) Contrast imaging demonstrating flow restoration after iliac vein stenting procedure, with two IVUS measurement points labeled b and c along the vessel. (**b**,**c**) Cross-sectional IVUS scans show lumen restoration and vessel patency.

**Table 1 diagnostics-15-00577-t001:** Comparative analysis of the main international guidelines about acute and chronic DVT management.

Guideline	Anticoagulation	Endovascular Treatment	Role of IVUS
**ESVS (2021)**	DOACS preferred, long-term therapy based on risk factors	Recommends CDT for iliofemoral DVT, venous stenting in chronic venous disease	Recognized IVUS for anatomical assessment but not mandatory **(Suggested)**
**ACCP (2021)**	DOACs preferred for most cases; 3-month treatment phase for all patients, extended therapy for unprovoked VTE or persistent risk factors	Recommends endovascular treatment in selected cases of severe iliofemoral obstruction	IVUS suggested for procedural planning and stent optimization **(Suggested)**
**SIR (2022)**	DOACs or LMWH, individualized duration	Strong recommendation for CDT, MT, and stenting in iliofemoral DVT	Strong emphasis on IVUS for procedural optimization **(Recommended)**
**AWMF (2024)**	DOACs preferred; anticoagulation 3–6 months for most cases	Endovascular therapy for high-risk cases, conservative for others	IVUS suggested for evaluating venous flow and obstruction **(Suggested)**

**Table 2 diagnostics-15-00577-t002:** Comparison of IVUS findings in acute DVT and PTS.

Feature	Acute DVT	PTS
**Thrombus appearance**	Hypoechoic, soft edges	Hyperechoic, firm, defined
**Vein wall**	Non-thickened, distended	Thickened, scarred
**Collaterals**	Minimal or absent	Extensive
**Valvular Function**	Frequently compromised	Compromised
**Luminal Obstruction**	Complete or partial	Partial, with narrowing

**Table 3 diagnostics-15-00577-t003:** Summary comparison of IVUS with other imaging modalities.

Imaging Modality	Sensitivity	Specificity	Advantages	Disadvantages
**IVUS**	**92–98%**	**95–99%**	Real-time imaging, high-resolution vein wall visualization, procedural guidance, no radiation	Invasive, costly, requires specialized training
**Doppler US**	**80–95%**	**85–98%**	Non-invasive, widely available, inexpensive	Limited in deep veins, poor in obese patients or with post-thrombotic changes
**Venography**	**90–96%**	**87–95%**	Gold standard for venous imaging, good for acute thrombosis	Invasive, uses contrast agents, radiation exposure, limited vein wall assessment
**CT Venography**	**85–97%**	**90–96%**	Excellent anatomical visualization, fast, useful for iliac and pelvic veins	Radiation exposure, contrast nephrotoxicity, limited assessment of intraluminal thrombus
**MR Venography**	**85–95%**	**90–98%**	No radiation, good for non-contrast evaluation, useful for central vein	Expensive, motion artifacts, less availability, longer scan time

**Table 4 diagnostics-15-00577-t004:** Role of IVUS in procedural guidance.

Procedure	IVUS Advantages	Alternative Without IVUS	Risk Without IVUS
**Catheter-directed Thrombolysis (CDT)**	Identifies thrombus burden and response to therapy	Venography	Inaccurate thrombus localization, higher drug dosage required
**Mechanical Thrombectomy**	Differentiates between soft and organized thrombi	Venography	Risk of incomplete thrombus removal
**Venous Stenting**	Measures precise vein diameter, confirms stent expansion	Venography + Doppler US	Undersized stents, risk of malpositioning
**Post-procedural assessment**	Confirms stent integrity and venous flow	Venography	Delayed restenosis or stent migration detection

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
