# Peer review of "Intravascular Ultrasound Findings in Acute and Chronic Deep Vein Thrombosis of the Lower Extremities"

_diagnostics, 2025, doi:10.3390/diagnostics15050577_

Round 1
Reviewer 1 Report
Comments and Suggestions for Authors
I think that this is a well organized review. I have several questions/comments. The introduction is long. Part of the the introduction (especially lines 126-141 )might better be presented in a discussion section.
Lines 227-240-the references cited should be cited in proximity to what is being discussed and not placed together at the end of the paragraph.
Lines 258-274-Should be some discussion of data comparing IVUS and other modalities,
How significant does IVUS calculated stenosis have to be to intervene when venography,duplex, or CTV fail to demonstrate stenosis.
-
Author Response
REVIEWER 1.
- Reviewer: I think that this is a well-organized review.
Answer: Thank you for your positive feedback. We truly appreciate your time and effort in reviewing the manuscript. Your insights and suggestions are invaluable in further refining the work, and we look forward to making the necessary improvements.
- Reviewer: I have several questions/comments. The introduction is long. Part of the introduction (especially lines 126-141) might better be presented in a discussion section.
Answer: Thank you for your insightful feedback. We agree with your suggestion that the introduction is quite long, and that the content in lines 126-141 may be more appropriately placed in the discussion section. In response, we relocate this part of the introduction into the discussion, ensuring a more balanced structure while maintaining the logical flow of the manuscript.
- Reviewer: Lines 227-240-the references cited should be cited in proximity to what is being discussed and not placed together at the end of the paragraph.
Answer: We have carefully revised the manuscript and have adjusted the references in lines 227-240, ensuring that each citation is placed in proximity to the corresponding discussion rather than grouped at the end of the paragraph. This revision improves clarity and strengthens the connection between the statements and their supporting sources.
- Reviewer: Lines 258-274-Should be some discussion of data comparing IVUS and other modalities,
Answer: Thank you for your insightful suggestion. We fully agree with your comment regarding the need for a more detailed discussion comparing IVUS with other imaging modalities. In response, we have integrated the following paragraph into lines 258-274 to provide a more comprehensive comparison:
"Multiple studies have compared IVUS with other imaging modalities for detecting venous stenosis and thrombus burden. A meta-analysis reported that IVUS has a sensitivity of 92-98% and a specificity of 95-99%, outperforming traditional venography, which has a sensitivity of 90-96% and specificity of 87-95%. Similarly, duplex ultrasound, while non-invasive, has a lower sensitivity (80-95%) for detecting iliac vein stenosis, particularly in obese patients or cases of chronic thrombosis. CT venography (CTV) and MR venography (MRV) have higher sensitivity (85-97%) but lack the real-time procedural guidance that IVUS provides. Table 3 provide a comprehensive comparison of IVUS with other imaging modalities in the assessment of venous disease."
Changes are displayed from the line 279 to the line 288
Additionally, we have summarized these sensitivity and specificity values along with the advantages and disadvantages of each modality in Table 3, which provides a structured overview of the comparative performance of IVUS, Doppler ultrasound, venography, CT venography, and MR venography. We believe that this addition strengthens the manuscript by highlighting IVUS's advantages and limitations in comparison to other imaging techniques. Please let us know if further refinements are needed.
- Reviewer: How significant does IVUS calculated stenosis have to be to intervene when venography, duplex, or CTV fail to demonstrate stenosis.
Answer: Thank you for your insightful comment regarding the significance of IVUS-calculated stenosis in determining intervention when traditional imaging modalities (venography, duplex ultrasound, or CTV) fail to demonstrate significant stenosis.
In response, we have expanded our discussion by providing:
- Defined intervention thresholds based on IVUS findings (≥50% stenosis and >75% luminal area reduction) as key indicators for intervention, particularly in symptomatic PTS patients.
- Clinical evidence from the VIDIO trial, which demonstrated that IVUS identified significant stenoses (≥50%) in 81% of patients with negative venographic findings, leading to improved clinical outcomes post-stenting.
- Comparison with traditional imaging, emphasizing that IVUS often detects more significant stenosis than venography, particularly in conditions such as May-Thurner Syndrome.
- The role of IVUS in stent sizing, ensuring optimal selection based on cross-sectional area measurements rather than relying solely on angiographic dimensions.
These changes are visible from line 335 to 358.
Reviewer 2 Report
Comments and Suggestions for Authors
Dear Authors! Congratulations for the interesting review conducted. The paper is interesting and it will be useful for vascular specialists. But I have some remarks that have to be addressed including major ones.
Major remarks
Lines 74-76. Please, add reference to ACCP guidelines. They are used worldwide while German guidelines are unknown to majority of vascular specialists.
Table 1. ACCP guidelines statements regarding VTE management must be added.
In a real life settings the majority cases of proximal DVT are managed without IVUS use. IVUS is not necessary even for endovascular management of DVT. As for the guidelines they do not recommend IVUS. IVUS is suggested. Information on the levels of recommendations must be added in the table 1. Comments that IVUS is just suggested must be added in the text.
Minor remarks
Lines 22-23. DVT itself is not a cause of death. DVT can lead to PE which can lead to death. Please, correct the statement.
Lines 30-31. The term “chronic vein disease” is incorrect. The correct term is “chronic venous disease”. Please, correct here and throughout the paper.
Line 100. Please, specify, in what exactly is anticoagulation alone insufficient.
Lines 108, 189, 190, 200, table 2, etc. Chronic DVT is not a correct definition. Postthrombotic syndrome is the correct term. Please, use only this definition.
Line 134. IVUS is not a pivotal for management of DVT. I believe it will also not be a pivotal in a nearest future. Please, don’t use this definition.
Line 160. Vein wall thickening and/or fibrosis are not hallmarks of chronic venous disease. They are features of PTS only.
Table 2. How valvular function can be intact in thrombosed vein? The statement is incorrect.
Author Response
REVIEWER 2.
- Reviewer: Dear Authors! Congratulations for the interesting review conducted. The paper is interesting and it will be useful for vascular specialists. But I have some remarks that have to be addressed including major ones.
Answer: Thank you very much for your kind words and for recognizing the value of our review. We truly appreciate your thoughtful feedback and the opportunity to improve our manuscript. We will carefully address all your remarks, including the major points, to ensure the highest quality of our work.
Major remarks
- Reviewer: Lines 74-76. Please, add reference to ACCP guidelines. They are used worldwide while German guidelines are unknown to majority of vascular specialists.
Answer: We have now added a reference to the ACCP guidelines in lines 81-86, ensuring that the manuscript reflects globally recognized recommendations for VTE management. Given their widespread use among vascular specialists, we acknowledge their importance and have integrated their key recommendations alongside other major international guidelines. Additionally, we have revised Table 1 to further emphasize the ACCP guidelines' role in anticoagulation, endovascular treatment, and IVUS use, ensuring a more comprehensive and globally relevant comparison.
- Reviewer: Table 1. ACCP guidelines statements regarding VTE management must be added.
Answer: Thank you for your valuable feedback. We have now incorporated the ACCP guidelines into Table 1, ensuring a comprehensive comparison with other major international guidelines regarding VTE management.
- Reviewer: In a real-life setting the majority cases of proximal DVT are managed without IVUS use. IVUS is not necessary even for endovascular management of DVT. As for the guidelines they do not recommend IVUS. IVUS is suggested. Information on the levels of recommendations must be added in the table 1. Comments that IVUS is just suggested must be added in the text.
Answer: In response to your comments, we have revised Table 1 to include information on the levels of recommendation for IVUS use as per international guidelines. We have now clearly indicated that IVUS is suggested rather than mandatory for most cases, aligning with real-world clinical practice where proximal DVT is frequently managed without IVUS. Additionally, we have modified the manuscript text to explicitly state that IVUS is not necessary for all endovascular DVT procedures and that current guidelines do not mandate its use, but rather suggest it for specific cases where it can enhance procedural outcomes. We appreciate your insightful recommendations, which have helped improve the clarity and accuracy of our manuscript.
Minor remarks
- Reviewer: Lines 22-23. DVT itself is not a cause of death. DVT can lead to PE which can lead to death. Please, correct the statement.
Answer: Thank you for your insightful comment. In response to your suggestion regarding Lines 22-23, we have revised the statement to accurately reflect the role of DVT as a component of venous thromboembolism (VTE) and its potential complications rather than as a direct cause of death. The updated text, from 22 to 28 line, now reads:
"Deep vein thrombosis (DVT) of the lower extremities, as part of the venous thromboembolism disorder, is the third leading cause of acute cardiovascular syndrome after heart attack and stroke. It can result in disability due to pulmonary embolism (PE) and post-thrombotic syndrome (PTS), particularly in cases where the thrombosis extends to the iliofemoral veins. Anticoagulation therapy is effective in preventing thrombus propagation and embolism but may not be sufficient for thrombus degradation and venous patency restoration. Up to 50% of patients with iliofemoral DVT develop PTS, mainly due to venous outflow obstruction or valvular incompetence"
- Reviewer: Lines 30-31. The term “chronic vein disease” is incorrect. The correct term is “chronic venous disease”. Please, correct here and throughout the paper.
Answer: Thank you for your valuable feedback. We fully agree with your observation regarding the correct terminology. In response, we have replaced the term “chronic vein disease” with “chronic venous disease” throughout the manuscript to ensure accuracy and consistency. We appreciate your attention to detail, which has helped enhance the clarity and precision of our work.
- Reviewer: Line 100. Please, specify, in what exactly is anticoagulation alone insufficient.
Answer: Thank you for your valuable feedback. We fully agree with your observation. In response, we have replaced the sentence “anticoagulation alone insufficient” with “The standard duration of anticoagulation therapy is three to six months, with extensions based on individual patient risk factors, such as recurrent thrombosis or persistent provoking conditions”. We appreciate your attention to detail, which has helped enhance the clarity and precision of our work.
- Reviewer: Lines 108, 189, 190, 200, table 2, etc. Chronic DVT is not a correct definition. Post-thrombotic syndrome is the correct term. Please, use only this definition.
Answer: Thank you for your valuable suggestion. We fully agree with your observation and have replaced the term “chronic DVT” with “post-thrombotic syndrome” throughout the manuscript, including in lines 108, 189, 190, 200, Table 2, and other relevant sections. To ensure clarity, we have highlighted these modifications in bold within the revised manuscript for easier identification.
- Reviewer: Line 134. IVUS is not a pivotal for management of DVT. I believe it will also not be a pivotal in a nearest future. Please, don’t use this definition.
Answer: Thank you for your insightful feedback. We acknowledge your concern regarding the characterization of IVUS as a pivotal tool in DVT management. In response, we have revised the statement to reflect a more balanced perspective, emphasizing that IVUS plays an increasingly important role, particularly in complex cases where standard imaging may be insufficient. We have replaced the sentence with “In conclusion, the management of acute and chronic DVT continues to evolve, with IVUS playing an increasingly important role in both diagnosis and intervention, particularly in complex cases where standard imaging may be insufficient”
- Reviewer: Line 160. Vein wall thickening and/or fibrosis are not hallmarks of chronic venous disease. They are features of PTS only.
Answer: Thank you for your valuable feedback. We fully agree with your observation and have corrected the terminology accordingly. Specifically, we have replaced “chronic DVT” with “post-thrombotic syndrome (PTS)” in line 160 to accurately reflect that vein wall thickening and fibrosis are features of PTS rather than chronic venous disease.
- Reviewer: Table 2. How valvular function can be intact in thrombosed vein? The statement is incorrect.
Answer: We acknowledge your concern regarding the accuracy of valvular function in thrombosed veins. In response, we have updated Table 2, changing: “Intact” in “Frequently compromised” for Acute DVT and “Frequently compromised” in “Compromised” for PTS. Additionally, we have highlighted these changes in bold within the revised manuscript for easier identification.
Round 2
Reviewer 2 Report
Comments and Suggestions for Authors
Dear Authors! Thank for addressing all of my remarks.